# The Personalized Nutrition Study (POINTS): evaluation of a genetically informed weight loss approach, a Randomized Clinical Trial

Christoph Höchsmann [1,2] ✉, Shengping Yang[2], José M. Ordovás [3], James L. Dorling [4], Catherine M. Champagne [2], John W. Apolzan [2], Frank L. Greenway [2], Michelle I. Cardel[5,6], Gary D. Foster[5,7] & Corby K. Martin [2]

Weight loss (WL) differences between isocaloric high-carbohydrate and high-fat diets are generally small; however, individual WL varies within diet groups. Genotype patterns may modify diet effects, with carbohydrate-responsive genotypes losing more weight on high-carbohydrate diets (and vice versa for fat-responsive genotypes). We investigated whether 12-week WL (kg, primary outcome) differs between genotype-concordant and genotype-discordant diets. In this 12-week single-center WL trial, 145 participants with overweight/obesity were identified a priori as fat-responders or carbohydrate-responders based on their combined genotypes at ten genetic variants and randomized to a high-fat (n = 73) or high-carbohydrate diet (n = 72), yielding 4 groups: (1) fat-responders receiving high-fat diet, (2) fat-responders receiving high-carbohydrate diet, (3) carbohydrate-responders receiving high-fat diet, (4) carbohydrate-responders receiving high-carbohydrate diet. Dietitians delivered the WL intervention via 12 weekly diet-specific small group sessions. Outcome assessors were blind to diet assignment and genotype patterns. We included 122 participants (54.4 [SD:13.2] years, BMI 34.9 [SD:5.1] kg/m², 84% women) in the analyses. Twelve-week WL did not differ between the genotype-concordant (−5.3 kg [SD:1.0]) and genotype-discordant diets (−4.8 kg [SD:1.1]; adjusted difference: −0.6 kg [95% CI: −2.1,0.9], p = 0.50). With the current ability to genotype participants as fat- or carbohydrate-responders, evidence does not support greater WL on genotype-concordant diets. ClinicalTrials identifier: NCT04145466.

The 2017–2018 National Health and Nutrition Examination Survey (NHANES) showed that almost 43% of US adults aged 20 and over have obesity, including 9.0% with severe obesity, and another 31% are overweight[1]. Excess body fat increases the risk of numerous medical conditions and premature mortality[2], presenting public health and economic challenges[3,4].

Many weight loss (WL) strategies emphasize either high-carbohydrate (and low-fat) or high-fat (low-carbohydrate) diets[5,6]. WL

[1]Department of Health and Sport Sciences, TUM School of Medicine and Health, Technical University of Munich, Munich, Germany. [2]Pennington Biomedical Research Center, Baton Rouge, LA, USA. [3]Tufts University, Boston, MA, USA. [4]Human Nutrition, School of Medicine, Dentistry and Nursing, College of Medical, Veterinary and Life of Sciences, University of Glasgow, Glasgow, UK. [5]WW International, Inc., New York, NY, USA. [6]Department of Health Outcomes and Biomedical Informatics, University of Florida College of Medicine, Gainesville, FL, USA. [7]Center for Weight and Eating Disorders, Perelman School of Medicine, University of Pennsylvania, Philadelphia, PA, USA. ✉e-mail: christoph.hoechsmann@tum.de

differences between isocaloric high-carbohydrate and high-fat diets are generally small or negligible[7]; however, individual WL varies substantially within diet groups[6], suggesting that individuals react differently to high-carbohydrate or high-fat diets. Retrospective data suggest that participants with carbohydrate-responsive polymorphisms lose more weight on high-carbohydrate vs. high-fat diets and vice versa for those with fat-responsive polymorphisms[8]. However, these results have not been confirmed in randomized controlled trials (RCT), and the approach of determining low-fat- and low-carbohydrate-responsive genotypes based on single-nucleotide polymorphisms (SNPs) from three genes (*PPARG*, *ADRB2*, and *FABP2*)[8,9] has been criticized[10]. Overall, reports show that most genotype × diet interactions are not significant, and replication is rare[11]. A more comprehensive and informative risk score (determined a priori), comprised of a greater number of SNPs with demonstrated and validated effects on the responses to high-fat/

high-carbohydrate diets, may better define fat- and carbohydrate-responsive genetic predisposition scores.

The present RCT tested the hypothesis that participants assigned to a diet corresponding to their a priori-determined (fat-responsive or carbohydrate-responsive) genotype would lose more weight over 12 weeks than those assigned to a diet discordant with their genotype. Further, we aimed to analyze those with a fat-responsive genotype (subsequently "fat-responders") and carbohydrate-responsive genotype (subsequently "carbohydrate-responders") separately. We hypothesized that (1) fat-responders would lose more weight on the high-fat vs. high-carbohydrate diet and conversely (2) carbohydrate-responders would lose more weight on the high-carbohydrate vs. high-fat diet. A secondary objective of the present RCT was to test the newly-developed genetic risk score to determine fat- and carbohydrate-responsive genotypes that was based on the current

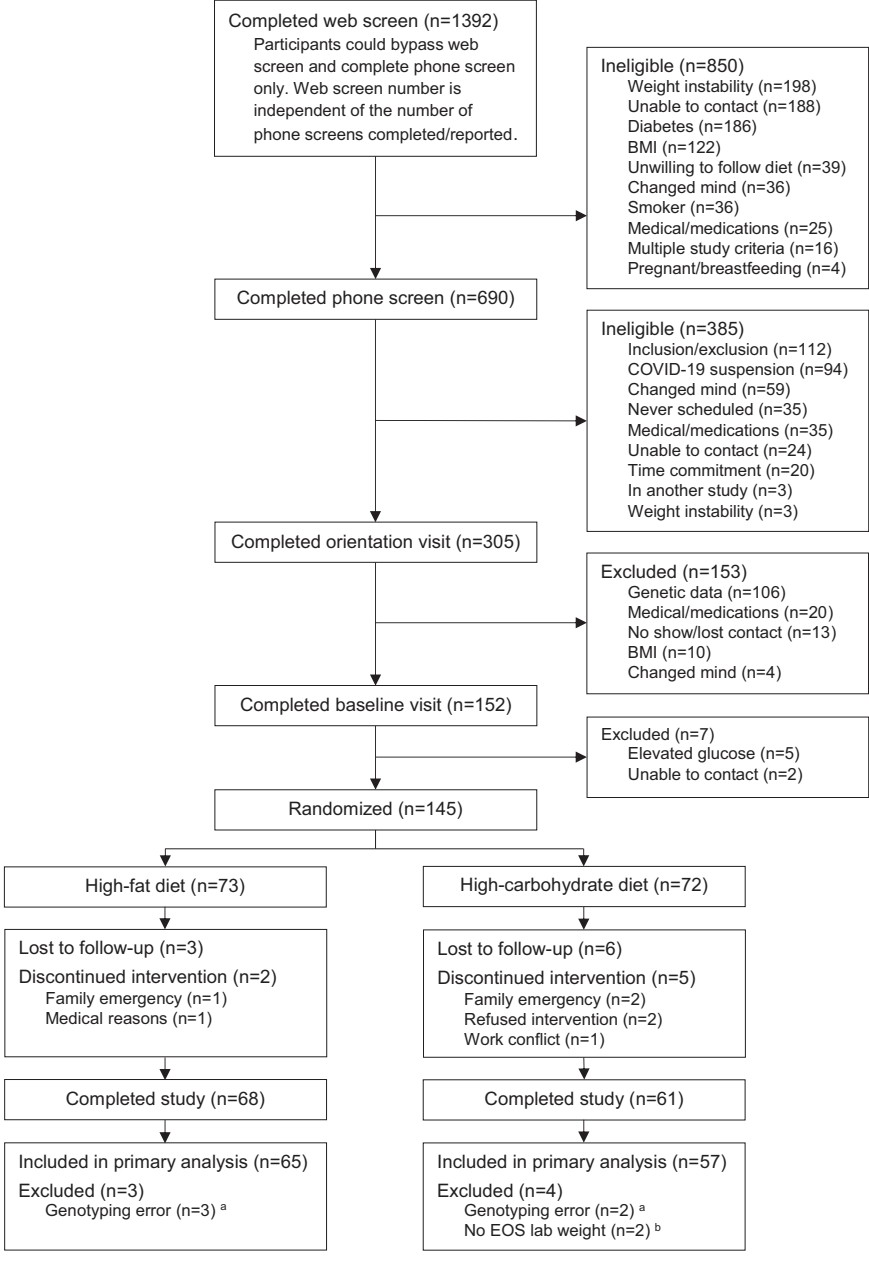

**Fig. 1 | CONSORT diagram illustrating the flow of participants through the POINTS trial.** [a]An error in the algorithm to determine carbohydrate- and fat-responsive genotypes led to the incorrect classification of these participants. These participants were erroneously enrolled as they did not meet the eligibility criteria.

This was reported to the IRB, and, as part of the resolution, their data were removed from the dataset. [b]These participants were unable to attend the W12 visit in person and only completed surveys and questionnaires remotely.

state-of-the-art in nutrigenomics. We also aimed to determine associations between baseline insulin levels and homeostatic model assessment for insulin resistance (HOMA-IR) and differential WL between the diets. These analyses were pursued as previous results were mixed with some studies finding that insulin resistance[12,13] and glucose-stimulated insulin secretion[14] influenced differential weight loss between low-fat and low-carbohydrate diets. In contrast, others found no interaction between glucose-stimulated insulin secretion and diet type on 12-month weight loss[9]. Finally, we examined the diet effects on eating attitudes and behaviors to help elucidate the mechanisms by which any observed differences in WL occurred. As program adherence diminishes over time[15], we chose a 12-week intervention period, which generally has lower attrition (~19%) than 6- (~35%) and 12-month (~54%) programs[16], and short-term WL is associated with long-term results[17,18].

## Results

Figure 1 shows the flow of participants through the study. Of the 2082 participants who screened for the study, 305 were eligible following the web/phone screen and were invited to the orientation visit. After eligibility verification based on medical history, medication inventory, and physical measures, 275 remained and completed a genealogy test. Of these 275 individuals, 106 (~39%) were excluded because they had a genotype that was classified as responsive to neither a high-fat nor a high-carbohydrate diet or as responsive to both diets. Of the remaining 169 individuals, 112 (~41%) were fat-responders, and 57 (~20%) were carbohydrate-responders. Before the baseline visit (completed by 152 participants), 17 participants were excluded because we either lost contact between the orientation visit and the baseline visit ($n = 13$) or because participants changed their minds about willingness to participate ($n = 4$). Following the baseline visit, 7 additional participants were excluded due to elevated glucose levels ($n = 5$) or lost contact ($n = 2$). Of the 145 participants randomized, 16 were lost to follow-up (W12), and 129 completed the trial. Seven participants were excluded from the analyses because they were incorrectly genotyped and erroneously enrolled ($n = 5$; removal from dataset suggested by IRB) or failed to provide weight data at W12 ($n = 2$). Baseline characteristics of all 122 included participants

(54.4 [SD: 13.2] years, BMI 34.9 [SD: 5.1] kg/m$^2$, 84% women, 68% White) are provided in Table 1. A comparison of baseline characteristics between non-completers ($n = 16$) and completers ($n = 122$) is provided in Supplementary Table 3.

### Change in the primary outcome

Weight change did not differ between genotype-concordant (−5.3 kg [SD: 1.0]) and genotype-discordant diets (−4.8 kg [SD: 1.1]; adjusted difference: −0.6 kg [95% CI: −2.1, 0.9, $p = 0.50$]; Table 2, Fig. 2). Among fat-responders, weight change did not differ between the high-fat (−5.5 kg [SD: 1.2]) and the high-carbohydrate diet (−5.3 kg [SD: 1.3]; adjusted difference: −0.2 kg [95% CI: −2.1, 1.6, $p = 0.78$]; Table 2). Similarly, among carbohydrate-responders, weight change did not differ between the high-carbohydrate (−5.1 kg [SD: 1.6]) and high-fat diet (−4.1 kg [SD: 1.7]; adjusted difference: −1.3 kg [95% CI: −3.9, 1.3, $p = 0.49$]; Table 2). Raw differences are presented in Supplementary Table 5.

### Percent weight change and change in body fat and body composition

Similar to absolute weight change, percent weight change (adjusted difference: −0.6% [95% CI: −2.1, 0.9, $p = 0.61$]) and change in body fat (adjusted difference: −0.5% [95% CI: −2.4, 1.4]) did not differ between genotype-concordant and genotype-discordant diets (Table 2, Fig. 2). Among fat-responders, percent weight change (adjusted difference: −0.2% [95% CI: −2.1, 1.7, $p = 0.83$]) and change in body fat (adjusted difference: 0.9% [95% CI: −1.3, 3.0]) did not differ between the high-fat and the high-carbohydrate diet (Table 2). Similarly, among carbohydrate-responders, percent weight change (adjusted difference: −1.2% [95% CI: −4.2, 1.7, $p = 0.57$]) and change in body fat (adjusted difference: −3.4% [95% CI: −7.5, 0.8]) did not differ between the high-carbohydrate and high-fat diet (Table 2). Changes in waist circumference (adjusted difference: −0.5 cm [95% CI: −2.3, 1.3]), hip circumference (adjusted difference: −1.0 cm [95% CI: −3.6, 1.6]), and waist-hip ratio (adjusted difference: 0.00 [95% CI: −0.02, 0.03]) did not differ between genotype-concordant and genotype-discordant diets (Table 2). Raw differences are presented in Supplementary Table 5.

## Table 1 | Participant characteristics

| | All participants (N = 122) | Fat-responders (n = 85) | | Carbohydrate-responders (n = 37) | |
| --- | --- | --- | --- | --- | --- |
| | | High-fat diet (n = 44) | High-carbohydrate diet (n = 41) | High-fat diet (n = 21) | High-carbohydrate diet (n = 16) |
| **Race, n (%)** | | | | | |
| White | 83 (68.0) | 30 (68.2) | 31 (75.6) | 12 (57.1) | 10 (62.5) |
| Black/ African American | 36 (29.5) | 12 (27.3) | 10 (24.4) | 9 (42.9) | 5 (31.2) |
| Other | 3 (2.5) | 2 (4.5) | 0 (0.0) | 0 (0.0) | 1 (6.2) |
| **Sex, n (%)** | | | | | |
| Female | 102 (83.6) | 37 (84.1) | 35 (85.4) | 17 (81.0) | 13 (81.2) |
| Male | 20 (16.4) | 7 (15.9) | 6 (14.6) | 4 (19.0) | 3 (18.8) |
| | Mean (SD) | Mean (SD) | Mean (SD) | Mean (SD) | Mean (SD) |
| Age, years | 54.4 (13.2) | 57.4 (11.5) | 54.4 (14.2) | 49.8 (14.1) | 52.4 (13.0) |
| Weight, kg | 94.3 (15.2) | 94.2 (14.0) | 93.5 (14.4) | 95.2 (17.6) | 95.6 (18.1) |
| BMI, kg/m$^2$ | 34.9 (5.1) | 35.1 (5.0) | 34.3 (4.8) | 35.8 (5.8) | 34.8 (5.3) |
| Body fat, % | 45.1 (9.3) | 45.0 (9.4) | 45.2 (8.5) | 43.8 (11.6) | 46.1 (8.4) |
| Waist circumference, cm | 109.0 (12.2) | 109.3 (11.8) | 108.5 (12.3) | 109.3 (11.8) | 109.2 (14.8) |
| Hip circumference, cm | 118.9 (12.2) | 117.5 (10.7) | 118.3 (12.1) | 120.1 (12.5) | 122.8 (15.8) |
| Waist-hip ratio | 0.92 (0.08) | 0.94 (0.09) | 0.92 (0.08) | 0.91 (0.06) | 0.89 (0.10) |
| SBP, mmHg | 121.7 (11.9) | 120.5 (11.6) | 124.1 (12.9) | 121.7 (11.5) | 119.4 (10.5) |
| DBP, mmHg | 74.7 (7.4) | 75.1 (7.0) | 74.5 (7.9) | 75.0 (6.1) | 73.9 (9.3) |

Data are mean (SD) for continuous and n (%) for categorical variables.
*BMI* body mass index, *DBP* diastolic blood pressure, *SBP* systolic blood pressure, *SD* standard deviation.

**Table 2 | Change in weight (kg and %), percent body fat, body composition, and blood pressure during the 12-week intervention in those assigned to a diet concordant vs. discordant with the genotype**

| All participants | Genotype-concordant diet (n = 60) Mean (SD) | Genotype-discordant diet (n = 62) Mean (SD) | Adjusted difference[a] (95% CI) | p-value |
|---|---|---|---|---|
| Weight change, kg | −5.3 (1.0) | −4.8 (1.1) | −0.6 (−2.1, 0.9) | 0.501 |
| Weight change, % | −5.8 (1.0) | −5.4 (1.1) | −0.6 (−2.1, 1.0) | 0.605 |
| Change in body fat, %[b] | −1.3 (1.2) | −0.8 (1.3) | −0.5 (−2.4, 1.4) | |
| Waist circumference, cm | −4.8 (1.1) | −4.3 (1.2) | −0.5 (−2.3, 1.3) | |
| Hip circumference, cm | −4.6 (1.7) | −3.7 (1.8) | −1.0 (−3.6, 1.6) | |
| Waist-hip ratio | 0.01 (0.00) | 0.01 (0.00) | 0.00 (−0.02, 0.03) | |
| SBP, mmHg | 1.2 (2.7) | −2.9 (2.9) | 4.7 (0.5, 8.8) | |
| DBP, mmHg | 0.4 (1.7) | 1.0 (1.9) | −0.1 (−2.8, 2.5) | |
| **Fat-responders** | **High-fat diet (n = 44) Mean (SD)** | **High-carbohydrate diet (n = 41) Mean (SD)** | **Adjusted difference[a] (95% CI)** | **p-value** |
| Weight change, kg | −5.5 (1.2) | −5.3 (1.3) | −0.2 (−2.1, 1.6) | 0.779 |
| Weight change, % | −5.9 (1.3) | −5.7 (1.4) | −0.2 (−2.1, 1.7) | 0.831 |
| Change in body fat, %[c] | −1.1 (1.4) | −1.9 (1.6) | 0.9 (−1.3, 3.0) | |
| Waist circumference, cm | −5.0 (1.4) | −4.4 (1.5) | −0.6 (−2.7, 1.5) | |
| Hip circumference, cm | −3.9 (1.5) | −4.0 (1.7) | 0.2 (−2.1, 2.6) | |
| Waist-hip ratio | 0.00 (0.00) | 0.00 (0.00) | −0.01 (−0.03, 0.02) | |
| SBP, mmHg | 4.5 (3.2) | −1.2 (3.5) | 6.9 (2.0, 11.8) | |
| DBP, mmHg | 1.7 (2.2) | 2.9 (2.4) | −0.5 (−3.8, 2.9) | |
| **Carbohydrate-responders** | **High-carbohydrate diet (n = 16)** **Mean (SD)** | **High-fat diet (n = 21)** **Mean (SD)** | **Adjusted difference[a] (95% CI)** | **p-value** |
| Weight change, kg | −5.1 (1.6) | −4.1 (1.7) | −1.3 (−3.9, 1.3) | 0.487 |
| Weight change, % | −5.7 (1.8) | −4.8 (1.9) | −1.2 (−4.2, 1.7) | 0.565 |
| Change in body fat, %[d] | −1.9 (2.5) | 1.4 (2.7) | −3.4 (−7.5, 0.8) | |
| Waist circumference, cm | −4.4 (2.1) | −4.2 (2.3) | −0.3 (−3.9, 3.3) | |
| Hip circumference, cm | −6.4 (4.3) | −2.7 (4.7) | −3.9 (−11.1, 3.3) | |
| Waist-hip ratio | 0.00 (0.00) | 0.00 (0.00) | 0.03 (−0.02, 0.08) | |
| SBP, mmHg | −5.8 (4.6) | −7.2 (5.0) | 0.3 (−7.4, 8.0) | |
| DBP, mmHg | −2.0 (2.8) | −3.0 (3.0) | 0.9 (−3.7, 5.5) | |

*CI* confidence interval, *DBP* diastolic blood pressure, *SBP* systolic blood pressure, *SD* standard deviation.
[a]Mixed-effect model, adjusted for sex, race, and baseline value of the outcome for all data.
[b]Data available for 58 of 60 participants (genotype-concordant diet) and 60 of 62 participants (genotype-discordant diet).
[c]Data available for 42 of 44 participants (high-fat diet) and 40 of 41 participants (high-carbohydrate diet).
[d]Data available for 16 of 16 participants (high-carbohydrate diet) and 20 of 21 participants (high-fat diet).

**Change in blood pressure**
Changes in resting systolic blood pressure (SBP) and DBP did not differ between genotype-concordant and genotype-discordant diets (SBP adjusted difference: 4.7 mmHg [95% CI: 0.5, 8.8]; DBP adjusted difference: −0.1 mmHg [95% CI: −2.8, 2.5]; Table 2, Fig. 3). Similarly, changes in SBP and DBP did not differ between the high-fat and the high-carbohydrate diet among fat-responders (SBP difference: 6.9 mmHg [95% CI: 2.0, 11.8]; DBP difference: −0.5 mmHg [95% CI: −3.8, 2.9]) or between the high-carbohydrate and high-fat diet among carbohydrate responders (SBP difference: 0.3 mmHg [95% CI: −7.4, 8.0]; DBP difference: 0.9 mmHg [95% CI: −3.7, 5.5]; Table 2). Raw differences are presented in Supplementary Table 5.

**Association between insulin levels and HOMA-IR and weight loss**
Baseline insulin levels ($\beta = −0.036$ [95% CI: −0.125, 0.053, $p = 0.43$]) and HOMA-IR ($\beta = −0.165$ [95% CI: −0.505, 0.175, $p = 0.34$]) were not associated with weight change (Supplementary Figure 1). There was no diet × baseline HOMA-IR interaction on weight change ($p = 0.37$). Similarly, there was no significant diet × baseline HOMA-IR interaction among carbohydrate-responders ($p = 0.62$) or fat-responders ($p = 0.23$; Supplementary Fig. 2).

**Change in food cravings, appetitive traits, and food preferences**
Changes in food cravings did not differ between the genotype-concordant and genotype-discordant diets (Table 3). Among carbohydrate-responders, those on a high-fat diet decreased cravings for carbohydrates/starches relative to those on the high-carbohydrate diet with an adjusted difference of −0.7 (95% CI: −1.1, −0.4, $p = 0.006$, without Holm-Bonferroni adjustment $p = 0.001$). Changes in all other food cravings did not differ between diets among carbohydrate-responders (Table 3). Among fat-responders, changes in food cravings did not differ between diets (Table 3). Raw differences are presented in Supplementary Table 6. Changes in restraint, disinhibition, and hunger (via EI), and food preferences (FPQ) did not differ between genotype-concordant and genotype-discordant diets (Table 4). Raw differences are presented in Supplementary Table 7 and baseline scores in these instruments are reported in Supplementary Table 4.

**Diet personalization and intervention satisfaction**
Diet preference (via Diet Personalization Survey, Table 5) and intervention satisfaction (Table 6) did not differ between the genotype-concordant and genotype-discordant diets. Raw differences are presented in Supplementary Table 8.

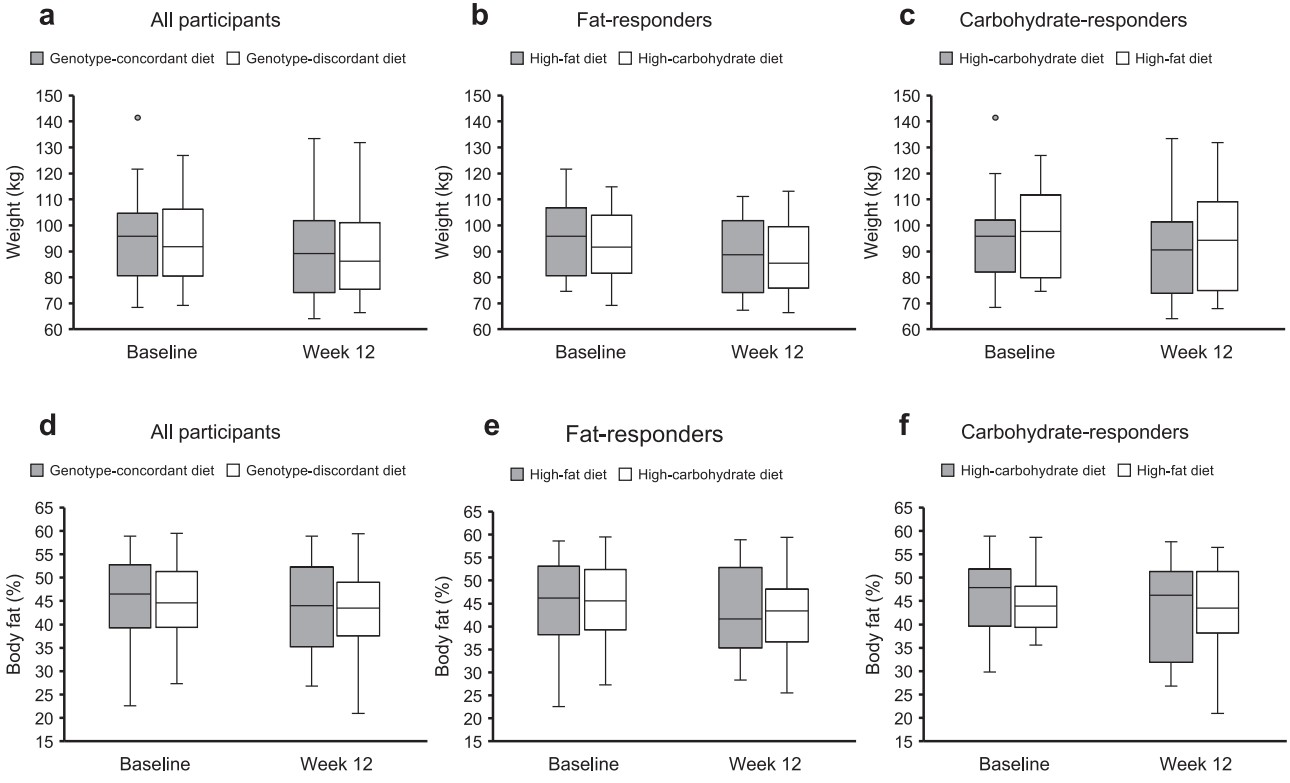

**Fig. 2 | Change in weight and percent body fat during the 12-week intervention.** Results are presented as boxplots for all participants (**a**, **d**), as well as for fat-responders (**b**, **e**) and carbohydrate responders (**c**, **f**) separately. **a** Genotype-concordant group ($n = 60$, genotype-discordant group ($n = 62$); **b** high-fat diet ($n = 44$), high-carbohydrate diet ($n = 41$); **c** high-carbohydrate diet ($n = 16$), high-fat diet ($n = 21$); **d** genotype-concordant group ($n = 58$), genotype-discordant group ($n = 60$); **e** high-fat diet ($n = 42$), high-carbohydrate diet ($n = 40$); **f** high-carbohydrate diet ($n = 16$), high-fat diet ($n = 20$). In the boxplots, the center line denotes the median value (50th percentile), the bounds of the box represent the 25th and 75th percentiles of the dataset, and the whiskers mark the 5th and 95th percentiles.

## Diet adherence

Adherence to the assigned diets is shown in Fig. 4. We encountered difficulties in obtaining the adherence data from participants due, in part, to the pandemic and needing to move to remote intervention delivery. Consequently, these adherence data are only available for 22 of 57 participants (39%) on the high-carbohydrate diet and for 43 of 65 participants (66%) on the high-fat diet (the discrepancy in the percent complete/missing is noted, though we have no reason to believe that it was systematic). On average, participants on the high-carbohydrate diet reported consuming 63.4% (SD: 2.3) of their energy from carbohydrates (target 65%), 20.9% (SD: 2.4) from fat (target 20%), and 16.0% (SD: 1.0) from protein (target 15%) in week 4, 63.3% (SD: 2.8) from carbohydrates, 20.5% (SD: 1.7) from fat, and 15.9% (SD: 1.0) from protein in week 8, and 62.7% (SD: 4.0) from carbohydrates, 20.5% (SD: 2.5) from fat, and 15.7% (SD: 1.8) from protein in week 12. Participants on the high-fat diet reported consuming on average 45.4% (SD: 2.2) of their energy from carbohydrates (target 45%), 39.4% (SD: 2.0) from fat (target 40%), and 15.8% (SD: 1.2) from protein (target 15%) in week 4, 44.7% (SD: 2.2) from carbohydrates, 40.5% (SD: 2.1) from fat, and 15.7% (SD: 2.3) from protein in week 8, and 44.5% (SD: 3.4) from carbohydrates, 39.9% (SD: 2.5) from fat, and 16.1% (SD: 3.3) from protein in week 12.

## Session attendance and adverse events

Weekly attendance was similar across the four genotype-diet groups (Supplementary Table 9), with weekly session attendance ranging from 85% to 100%. There were 4 adverse or serious adverse events in total. Two adverse events occurred among fat-responders on a high-carbohydrate diet (unrelated to the study), and there were 2 serious adverse events (1 among fat-responders on a high-carbohydrate diet, 1

among fat-responders on a high-fat diet) that required hospitalization (unrelated to study).

## Discussion

The present RCT determined the participant's (fat-responsive or carbohydrate-responsive) genotype a priori via a comprehensive genetic risk score based on published and validated effects and tested the effects of a genotype-concordant diet on WL over 12 weeks. We found no difference in WL between individuals on the genotype-concordant vs. genotype-discordant diet. Further, insulin levels or HOMA-IR were not associated with WL. Food cravings tended to decrease among carbohydrate-responders on a high-fat diet compared to those on a high-carbohydrate diet. Finally, fat-responders on a high-carbohydrate diet tended to decrease resting SBP.

The lack of significant and clinically meaningful differences in WL (-0.6 kg) between genotype-concordant and genotype-discordant diets aligns with the literature[9,11]. In contrast to the well-conducted Gardner et al. study (non-significant difference in WL of 0.7 kg over 12 months)[9], who defined fat vs. carbohydrate-responsive genotypes based on 3 SNPs that were predictive in a preliminary retrospective analysis[8], we determined fat- or carbohydrate-responsive genotypes based on an algorithm involving 10 SNPs. Supported by a recent-meta-analysis (8 trials with 91 SNPs and 63 genetic loci)[11], our results suggest that with the current ability to genotype individuals as fat or carbohydrate-responders, there is no evidence that genotype-concordant diets result in greater WL.

Our sample consisted of substantially fewer carbohydrate-responders ($n = 37$) than fat-responders ($n = 85$). We did not limit recruitment to achieve equal numbers of participants in each genotype-diet group, and this distribution reflects the prevalence in our

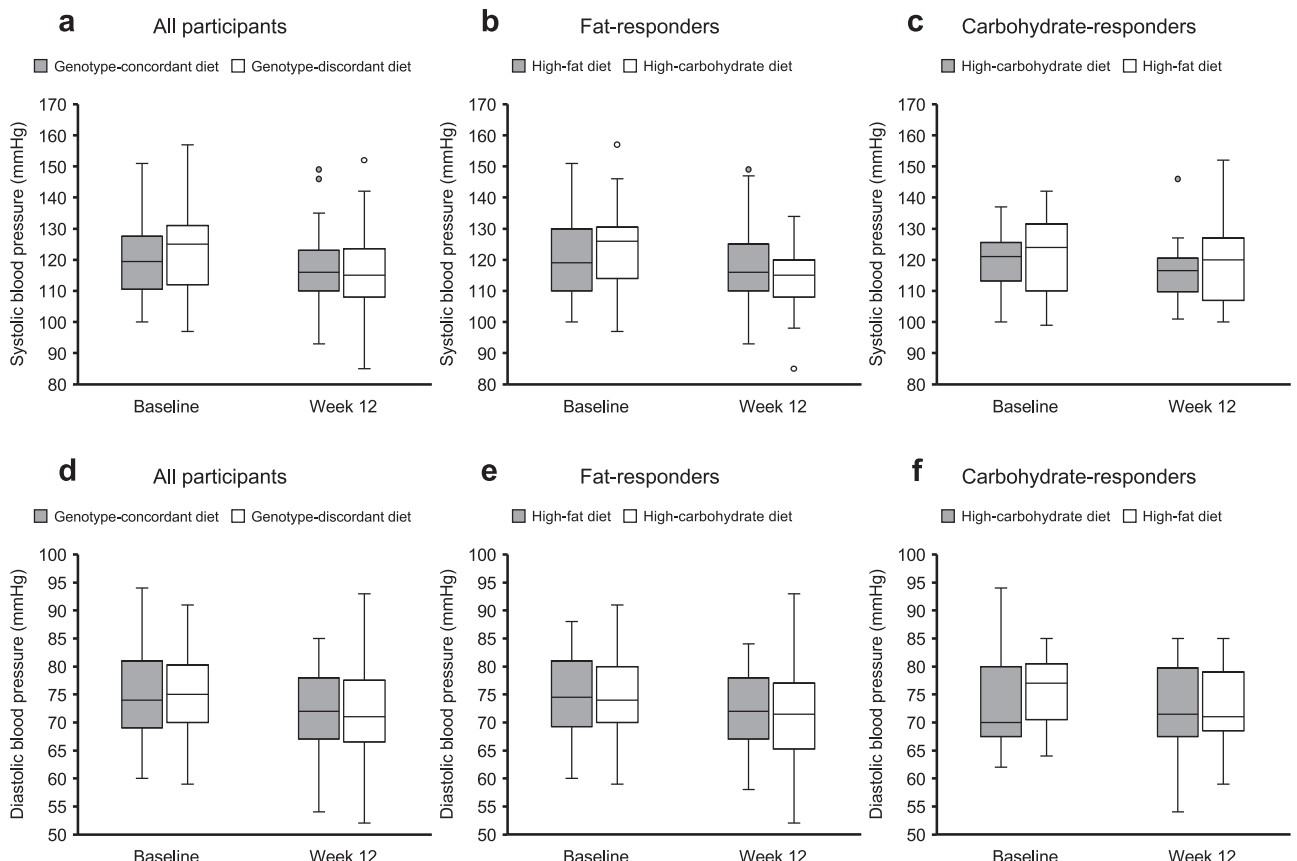

**Fig. 3 | Change in systolic and diastolic blood pressure during the 12-week intervention.** Results are presented as boxplots for all participants (**a**, **d**; genotype-concordant group, $n = 60$, genotype-discordant group, $n = 62$), as well as for fat-responders (**b**, **e**; high-fat diet, $n = 44$, high-carbohydrate diet, $n = 41$) and carbohydrate responders (**c**, **f**; high-carbohydrate diet, $n = 16$, high-fat diet, $n = 21$) separately. In the boxplots, the center line denotes the median value (50th percentile), the bounds of the box represent the 25th and 75th percentiles of the dataset, and the whiskers mark the 5th and 95th percentiles.

population. As reported in the Results section, 275 individuals completed a genealogy test, of which ~39% had a genotype classified as responsive to both or neither of the two diets, ~41% were fat-responders and ~20% were carbohydrate-responders. Notably, these numbers are somewhat different from what we had estimated during the study's planning phase, as we expected 1/3 of people to be fat-responders, 1/3 carbohydrate-responders, and 1/3 to respond to neither or both of the specified diets. Future studies with larger samples should verify if this uneven distribution between carbohydrate-responders and fat-responders is representative of the general population and further investigate the potential effect on WL among carbohydrate-responders.

Future studies could also consider assigning participants to genotype-concordant diets without specific energy intake targets and examine the diet effects not only on WL but also on cardiovascular risk factors. Previously, a low-carbohydrate diet without energy intake target resulted in greater improvements in body composition, blood lipids, and estimated 10-year coronary heart disease risk compared to a low-fat diet[19]. It would be insightful to investigate whether genotype plays a role in cardiovascular risk reduction following a low-carbohydrate vs. low-fat diet without calorie restriction.

Fasting insulin levels and HOMA-IR did not predict WL. Previous studies reporting a diet × fasting insulin interaction for WL found lower carbohydrate diets to be superior for individuals with greater insulin resistance[13] and high baseline insulin secretion (30 min after a 75 g oral glucose tolerance test)[20], presumably due to a reduced burden on insulin-mediated glucose disposal. However, these studies involved relatively small sample sizes, and findings of the influence of insulin sensitivity[21] and insulin secretion[9,14] on WL via a low-fat vs. a low-carbohydrate diet are inconsistent.

WL can reduce food cravings, particularly for foods restricted on specific diets[22], contributing to the hypothesis that food cravings are a conditioned expression of hunger due to stimuli paired with eating certain foods[23]. Consequently, cravings can be reduced by eliminating or restricting the intake of craved foods. This hypothesis is partially supported by our results as, among carbohydrate-responders, cravings tended to decrease for high-carbohydrate foods on the high-fat diet. Nonetheless, cravings also decreased modestly for high-fat foods, which is to be expected as the amount of all foods was restricted, and cravings for specific foods correlate with each other[24].

Among fat-responders, a high-carbohydrate diet tended to decrease resting SBP. Nonetheless, these individuals had the highest mean SBP of the 4 genotype-diet groups at baseline. Thus, this effect could be explained, in whole or partially, by regression to the mean. Also, all 4 genotype-diet groups had relatively well-controlled blood pressure, leaving little room for improvement through dietary changes, making the non-significant improvements potentially more meaningful.

This trial has some limitations. First, the genetic algorithm to classify individuals as fat- or carbohydrate-responders was created based on published literature[25–38]. However, these (mostly retrospective) studies generally had modest sample sizes, and some of the genotype × diet interactions, which may be false positives, have not been independently replicated. Further, WL is determined by multiple modifiable and non-modifiable (e.g., genetic) factors, and current knowledge accounts for a small percentage of the variability. Further genotypes may have influenced participants' WL responses in directions different from those predicted from the measured genotypes. More comprehensive knowledge of the role of

**Table 3 | Changes in food cravings (via the Food Craving Inventory) during the 12-week intervention in those assigned to a diet concordant vs. discordant with the genotype**

| All participants | Genotype-concordant diet (*n* = 60) Mean (SD) | Genotype-discordant diet (*n* = 62) Mean (SD) | Adjusted difference[a] (95% CI) |
|---|---|---|---|
| High fats[b] | −0.3 (0.1) | −0.4 (0.2) | 0.1 (−0.1, 0.4) |
| Sweets[c] | −0.3 (0.2) | −0.5 (0.2) | 0.2 (−0.1, 0.4) |
| Carbohydrates/Starches[d] | −0.1 (0.2) | −0.4 (0.2) | 0.3 (0.0, 0.5) |
| Fast-food fats[e] | −0.3 (0.2) | −0.4 (0.2) | 0.1 (−0.2, 0.4) |
| Fruits and vegetables[f] | −0.1 (0.2) | −0.4 (0.2) | 0.2 (−0.1, 0.5) |
| Total cravings[g] | −0.2 (0.1) | −0.4 (0.1) | 0.2 (−0.1, 0.4) |
| **Fat-responders** | **High-fat diet (*n* = 44) Mean (SD)** | **High-carbohydrate diet (*n* = 41) Mean (SD)** | **Adjusted difference[a] (95% CI)** |
| High fats[b] | −0.4 (0.2) | −0.3 (0.2) | −0.1 (−0.3, 0.3) |
| Sweets[c] | −0.4 (0.2) | −0.6 (0.2) | 0.2 (−0.1, 0.5) |
| Carbohydrates/Starches[d] | −0.2 (0.2) | −0.3 (0.2) | 0.1 (−0.3, 0.4) |
| Fast-food fats[e] | −0.4 (0.2) | −0.3 (0.3) | −0.1 (−0.4, 0.3) |
| Fruits and vegetables[f] | −0.3 (0.2) | −0.4 (0.3) | 0.1 (−0.3, 0.5) |
| Total cravings[g] | −0.3 (0.2) | −0.3 (0.2) | 0.0 (−0.3, 0.3) |
| **Carbohydrate-responders** | **High-carbohydrate diet (*n* = 16) Mean (SD)** | **High-fat diet (*n* = 21) Mean (SD)** | **Adjusted difference[a] (95% CI)** |
| High fats[b] | −0.2 (0.2) | −0.7 (0.2) | 0.5 (0.1, 0.9) |
| Sweets[c] | −0.1 (0.2) | −0.3 (0.3) | 0.2 (−0.2, 0.6) |
| Carbohydrates/Starches[d] | 0.1 (0.2) | −0.7 (0.2) | 0.7 (0.4, 1.1) |
| Fast-food fats[e] | −0.1 (0.3) | −0.5 (0.3) | 0.5 (0.0, 1.0) |
| Fruits and vegetables[f] | 0.3 (0.3) | −0.3 (0.3) | 0.6 (0.1, 1.1) |
| Total cravings[g] | 0.0 (0.2) | −0.5 (0.2) | 0.5 (0.2, 0.9) |

*CI* confidence interval, *SD* standard deviation.

[a]Adjusted for sex, race, and baseline value of the outcome.

[b]Genotype-concordant diet: 55/60 participants; genotype-discordant diet: 60/62 participants. Fat-responders: 41/44 participants (high-fat diet) and 40/41 participants (high-carbohydrate diet). Carbohydrate-responders: 14/16 participants (high-carbohydrate diet) and 20/21 participants (high-fat diet).

[c]Genotype-concordant diet: 59/60 participants; genotype-discordant diet: 60/62 participants. Fat-responders: 43/44 participants (high-fat diet) and 40/41 participants (high-carbohydrate diet). Carbohydrate-responders: 16/16 participants (high-carbohydrate diet) and 20/21 participants (high-fat diet).

[d]Genotype-concordant diet: 59/60 participants; genotype-discordant diet: 61/62 participants. Fat-responders: 44/44 participants (high-fat diet) and 40/41 participants (high-carbohydrate diet). Carbohydrate-responders: 15/16 participants (high-carbohydrate diet) and 21/21 participants (high-fat diet).

[e]Genotype-concordant diet: 58/60 participants; genotype-discordant diet: 61/62 participants. Fat-responders: 43/44 participants (high-fat diet) and 40/41 participants (high-carbohydrate diet). Carbohydrate-responders: 15/16 participants (high-carbohydrate diet) and 20/21 participants (high-fat diet).

[f]Genotype-concordant diet: 58/60 participants; genotype-discordant diet: 60/62 participants. Fat-responders: 43/44 participants (high-fat diet) and 40/41 participants (high-carbohydrate diet). Carbohydrate-responders: 15/16 participants (high-carbohydrate diet) and 20/21 participants (high-fat diet).

[g] Genotype-concordant diet: 54/60 participants; genotype-discordant diet: 67/62 participants. Fat-responders: 41/44 participants (high-fat diet) and 38/41 participants (high-carbohydrate diet). Carbohydrate-responders: 13/16 participants (high-carbohydrate diet) and 19/21 participants (high-fat diet).

genetics in WL is needed and should be obtained from genome-wide association studies; however, the sample size and experimental design required to generate that essential information are beyond reach at this time. Additional limitations of the present study include the relatively small sample size, single-center design, and short time frame. A longer timeframe (6–12-month follow-up) may have increased the amount and differential weight loss between diets. A larger sample size might have also allowed for detecting differences in clinically important secondary outcomes such as changes in body fat and SBP. Further, we did not provide meals in this study, which may have affected dietary adherence (high-fat vs. high-carbohydrate). However, this choice was made by design, as our study was designed as a (pragmatic) effectiveness trial with real-world conditions rather than an efficacy trial. Additionally, the adherence data (albeit limited) suggests that diet adherence was overall satisfactory. In addition to assessing diet adherence continuously throughout the study, future studies should also assess the macronutrient composition of participants' habitual diets to see any differences in the magnitude of the shifts from baseline to the high-fat or high-carbohydrate diet. Further, when assessing a potential effect modification by insulin resistance status, using an oral glucose tolerance test (AUC or INS-30) rather than HOMA-IR to quantify insulin resistance might have been a better option, as

HOMA-IR has limited sensitivity due to its reliance on fasting insulin and glucose levels and it does not reflect differences between tissues (e.g., adipose, muscle) or postprandial physiology. Non-fasting methods yield greater variability of the glucose/insulin dynamics and may have been more suitable. Additionally, the assessment of percent body fat via BIA is a limitation as BIA does not provide information on body fat distribution. Finally, participation in "nutri-genomics" studies generally induces improved diet adherence[39–42], independent of the specific recommendations. Therefore, in our study, participants may have responded better to their assigned diets regardless of their genotype matching, obscuring the specific nutri-genomics effects.

In conclusion, in this 12-week RCT, there was no difference in WL between individuals with an a priori determined fat- or carbohydrate-responsive genotype on a high-carbohydrate vs. high-fat diet with specific energy targets and the same level of energy restriction across diets.

## Methods
### Design and participants
The Personalized Nutrition Study (POINTS, ClinicalTrials.gov identifier: NCT04145466) was a 12-week, single-site, parallel-arm WL trial that was approved by the institutional review board (IRB FWA 00006218) of the

**Table 4 | Change in restraint, disinhibition, and hunger and in food preferences during the 12-week intervention in those assigned to a diet concordant vs. discordant with the genotype**

| All participants | Genotype-concordant diet (n = 60) Mean (SD) | Genotype-discordant diet (n = 62) Mean (SD) | Adjusted difference[a] (95% CI) |
|---|---|---|---|
| Restraint (EI)[b] | 3.6 (0.9) | 3.3 (1.0) | 0.4 (−1.1, 1.9) |
| Disinhibition (EI)[c] | −0.1 (0.6) | 0.1 (0.7) | 0.0 (−1.0, 0.9) |
| Hunger (EI)[d] | −0.4 (0.5) | −0.9 (0.6) | 0.5 (−0.4, 1.4) |
| HF/HS (FPQ) | −0.1 (0.3) | 0.0 (0.4) | 0.0 (−0.5, 0.5) |
| LF/HS (FPQ) | 0.1 (0.3) | 0.1 (0.3) | 0.1 (−0.4, 0.5) |
| HF/HCCHO (FPQ) | −0.3 (0.3) | −0.3 (0.3) | 0.0 (−0.4, 0.5) |
| LF/HCCHO (FPQ) | −0.1 (0.3) | 0.0 (0.3) | −0.1 (−0.5, 0.4) |
| HF/LCHO/HP (FPQ) | −0.4 (0.3) | −0.4 (0.3) | 0.0 (−0.5, 0.4) |
| LF/LCHO/HP (FPQ) | 0.1 (0.3) | 0.1 (0.3) | 0.0 (−0.4, 0.4) |
| **Fat-responders** | **High-fat diet (n = 44) Mean (SD)** | **High-carbohydrate diet (n = 41) Mean (SD)** | **Adjusted difference[a] (95% CI)** |
| Restraint (EI)[b] | 3.5 (1.2) | 2.7 (1.4) | 0.8 (−1.3, 2.9) |
| Disinhibition (EI)[c] | −0.3 (0.8) | 0.2 (0.9) | −0.4 (−1.6, 0.9) |
| Hunger (EI)[d] | −0.9 (0.7) | −1.3 (0.8) | 0.4 (−0.8, 1.5) |
| HF/HS (FPQ) | 0.0 (0.4) | 0.0 (0.5) | 0.1 (−0.6, 0.7) |
| LF/HS (FPQ) | 0.2 (0.4) | 0.3 (0.4) | 0.0 (−0.6, 0.5) |
| HF/HCCHO (FPQ) | −0.2 (0.4) | −0.1 (0.4) | 0.0 (−0.6, 0.5) |
| LF/HCCHO (FPQ) | 0.0 (0.4) | 0.2 (0.4) | −0.2 (−0.7, 0.4) |
| HF/LCHO/HP (FPQ) | −0.5 (0.4) | −0.6 (0.4) | 0.1 (−0.5, 0.7) |
| LF/LCHO/HP (FPQ) | 0.2 (0.4) | 0.1 (0.4) | 0.1 (−0.4, 0.7) |
| **Carbohydrate-responders** | **High-carbohydrate diet (n = 16) Mean (SD)** | **High-fat diet (n = 21) Mean (SD)** | **Adjusted difference[a] (95% CI)** |
| Restraint (EI)[b] | 3.4 (1.1) | 4.6 (1.1) | −0.7 (−2.6, 1.2) |
| Disinhibition (EI)[c] | 0.7 (0.9) | 0.0 (0.9) | 0.8 (−0.8, 2.5) |
| Hunger (EI)[d] | 0.8 (0.8) | −0.1 (0.9) | 1.0 (−0.4, 2.5) |
| HF/HS (FPQ) | −0.2 (0.5) | 0.0 (0.5) | −0.1 (−1.0, 0.7) |
| LF/HS (FPQ) | −0.1 (0.4) | −0.4 (0.4) | 0.3 (−0.3, 0.9) |
| HF/HCCHO (FPQ) | −0.4 (0.5) | −0.6 (0.5) | 0.2 (−0.6, 1.0) |
| LF/HCCHO (FPQ) | −0.2 (0.4) | −0.5 (0.4) | 0.3 (−0.3, 0.9) |
| HF/LCHO/HP (FPQ) | −0.3 (0.4) | 0.1 (0.5) | −0.2 (−1.0, 0.5) |
| LF/LCHO/HP (FPQ) | −0.1 (0.4) | 0.1 (0.4) | −0.2 (−0.8, 0.4) |

CI confidence interval, EI Eating Inventory, FPQ Food Preference Questionnaire, HF/HS high fat/high simple sugar, LF/HS low fat/high simple sugar, HF/HCCHO high fat/high complex carbohydrate, LF/HCCHO low fat/high complex carbohydrate, HF/LCHO/HP high fat/low carbohydrate/high protein, LF/LCHO/HP low fat/low carbohydrate/high protein, SD standard deviation.
[a]Adjusted for sex, race, and baseline value of the outcome.
[b]Genotype-concordant diet: 46/60 participants; genotype-discordant diet: 47/62 participants. Fat-responders: 34/44 participants (high-fat diet) and 29/41 participants (high-carbohydrate diet). Carbohydrate-responders: 12/16 participants (high-carbohydrate diet) and 18/21 participants (high-fat diet).
[c]Genotype-concordant diet: 49/60 participants; genotype-discordant diet: 49/62 participants. Fat-responders: 37/44 participants (high-fat diet) and 31/41 participants (high-carbohydrate diet). Carbohydrate-responders: 12/16 participants (high-carbohydrate diet) and 18/21 participants (high-fat diet).
[d]Genotype-concordant diet: 51/60 participants; genotype-discordant diet: 51/62 participants. Fat-responders: 37/44 participants (high-fat diet) and 33/41 participants (high-carbohydrate diet). Carbohydrate-responders: 14/16 participants (high-carbohydrate diet) and 18/21 participants (high-fat diet).

Pennington Biomedical Research Center (PBRC, Baton Rouge, LA). Participants were enrolled between October 7, 2020 and September 8, 2021. Participants were identified a priori as carbohydrate-responders and fat-responders based on their combined genotypes at 10 genetic variant loci and randomized to either a high-carbohydrate or high-fat diet, yielding the following groups: (1) fat-responders receiving a high-fat diet, (2) fat-responders receiving a high-carbohydrate diet, (3) carbohydrate-responders receiving a high-fat diet, and (4) carbohydrate-responders receiving a high-carbohydrate diet.

Participants were recruited from the community. Eligible participants were 18–75 years old, had a BMI of 27.0–47.5 kg/m², and had completed or were willing to complete a genealogy test (e.g., Ancestry, 23andMe) and to share the raw data with the investigators. Finally, a genetic profile indicating a predisposition to respond favorably to a high-carbohydrate or high-fat WL diet based on specific SNPs (see below) was required. Exclusion criteria included smoking, weight change ≥10 lbs. in the last 3 months, being pregnant or breastfeeding,

conditions, diseases, or medications that affect body weight or metabolism or could affect risk or study completion, and a genotype indicating a predisposition to respond favorably to neither or both of the specified diets. We estimated that approximately 1/3 of people would be fat-responders, 1/3 carbohydrate-responders, and 1/3 would respond favorably to neither or both of the specified diets.

The study included 1 orientation visit, 2 clinic visits (one before and one after the intervention), and 12-weekly intervention sessions. All participants provided written informed consent, and participants who completed the study received a minor compensation of $150.

**Genotype determination.** Carbohydrate- and fat-responders were identified a priori based on their combined genotypes at the following genetic variants: (1) FGF21rs838147[25], (2) TCF7L2rs12255372[26,43], (3) IRS1rs2943641[28], (4) APOA5rs662799[30,31,44], (5) PLIN1rs894160[27,32], (6) APOA2rs5082[29,33], (7) FTOrs9939609[34,35], (8) PPARGrs1801282[36], (9) GIPRrs10423928[37], and (10) GYS2rs1478290[38]. The genetic information

**Table 5 | Change in items of the Diet Personalization Survey during the 12-week intervention in those assigned to a diet concordant vs. discordant with the genotype**

| All participants | Genotype-concordant diet (n = 60) Mean[a] (SD) | Genotype-discordant diet (n = 62) Mean[a] (SD) | Adj. difference[b] (95% CI) |
|---|---|---|---|
| The assigned diet... | | | |
| ... fits my typical eating habits | 0.9 (0.5) | 1.3 (0.6) | −0.3 (−1.2, 0.6) |
| ... fits my lifestyle | 0.4 (0.6) | 0.2 (0.6) | 0.2 (−0.7, 1.1) |
| ... makes it easier to lose weight | 0.6 (0.5) | 0.7 (0.6) | 0.1 (−0.8, 0.8) |
| I am confident that I can... | | | |
| ... successfully lose weight on the assigned diet | 0.4 (0.5) | 0.6 (0.5) | −0.1 (−0.8, 0.7) |
| ... follow the assigned diet | −0.7 (0.4) | −0.4 (0.5) | −0.3 (−0.9, 0.4) |
| **Fat-responders** | **High-fat diet (n = 44) Mean[a] (SD)** | **High-carbohydrate diet (n = 41) Mean[a] (SD)** | **Adj. difference[b] (95% CI)** |
| The assigned diet... | | | |
| ... fits my typical eating habits | 0.6 (0.6) | 1.3 (0.7) | −0.6 (−1.6, 0.4) |
| ... fits my lifestyle | 0.0 (0.7) | 0.2 (0.8) | −0.1 (−1.2, 1.0) |
| ... makes it easier to lose weight | 0.2 (0.7) | 0.4 (0.7) | −0.1 (−1.1, 0.9) |
| I am confident that I can... | | | |
| ... successfully lose weight on the assigned diet | 0.1 (0.6) | 0.4 (0.7) | −0.1 (−1.1, 0.8) |
| ... follow the assigned diet | −0.9 (0.6) | −0.5 (0.6) | −0.4 (−1.2, 0.5) |
| **Carbohydrate-responders** | **High-carbohydrate diet (n = 16) Mean[a] (SD)** | **High-fat diet (n = 21) Mean[a] (SD)** | **Adj. difference[b] (95% CI)** |
| The assigned diet... | | | |
| ... fits my typical eating habits | 1.6 (1.1) | 1.3 (1.2) | 0.4 (−1.4, 2.2) |
| ... fits my lifestyle | 1.1 (1.0) | 0.3 (1.1) | 1.0 (−0.7, 2.7) |
| ... makes it easier to lose weight | 1.3 (0.7) | 1.3 (0.8) | 0.4 (−0.8, 1.6) |
| I am confident that I can... | | | |
| ... successfully lose weight on the assigned diet | 0.9 (0.8) | 1.1 (0.9) | 0.2 (−1.1, 1.5) |
| ... follow the assigned diet | −0.3 (0.7) | −0.2 (0.7) | 0.0 (−1.1, 1.1) |
| ... the degree to which the diet helped manage hunger | 6.8 (0.9) | 6.6 (1.0) | 0.3 (−1.3, 1.8) |

[a]Mean change during the 12-week intervention.
[b]Adjusted for sex and race.

was accessed via the raw data from the genealogy tests. Initially, only 6 SNPs were included and pilot tested, and the scoring criteria were then modified as few participants were deemed carbohydrate- or fat-responders. The original and updated scoring criteria, including a specific example for 1 SNP, are provided in the Supplementary Methods, including Supplementary Tables 1 and 2. The final risk score comprised 10 SNPs with demonstrated and validated effects on the responses to high-fat/high-carbohydrate diets[25–38,43,44], and validation of this comprehensive and informative risk score was an objective of this study.

**Intervention**
After enrollment (Week [W] 0 visit), participants were randomized to either a high-carbohydrate diet (rich in whole-grain foods) or a high-fat diet (rich in unsaturated fats/oils). The high-carbohydrate diet consisted of ~20% of energy from fat and ~65% from carbohydrates, whereas the high-fat diet consisted of ~40% energy from fat and ~45% from carbohydrates. Both diets provided 15% of energy from protein. All participants were assigned an energy intake target that would result in a daily deficit of ~750 kcal and provided with a diet-specific meal plan in 200 kcal increments from 1400 to 2800 kcal/day to self-prepare meals during the intervention period. To facilitate meal plan adherence when preparing or selecting meals, the meal plans included a list of ingredients (and their amounts) for all meals of each day (breakfast, lunch, dinner, and 1 daily snack) and instructions for meal preparation and participants were provided a food scale. Baseline energy requirements were calculated with Mifflin-St. Jeor's formulas[45].

The PBRC biostatistics department created the randomization sequence using SAS 9.4 statistical software for Windows (SAS Institute, Cary, NC) and uploaded it to REDCap (Research Electronic Data Capture). REDCap used strata for the inaction of genotype and gender. To ensure a relatively equal baseline BMI between the 4 genotype-diet groups, a 1:1 randomization scheme was devised that adjusted for BMI, gender, and genotype. Gender and genotype were used as strata, while BMI was used in an a-priori-created randomization equation. Within each stratum, this equation used block sizes of 6 (for females) and 4 (for males) at the start of the study and ended with block sizes of 4 and 2, respectively, to ensure relative balance of group assignments. Block sizes were assigned during the study by the biostatistician with access only to information about the enrolment progress (percent enrolled).

Outcome assessors were blind to diet assignment and genotype patterns. Interventionists administering intervention sessions were blind to genotype patterns but not diet type. Participants were only informed of their genotype (carbohydrate- or fat-responder) once they completed the study.

The 12 weekly intervention (group) sessions were diet-specific and had a different focus each week (Supplementary Material). Participants were provided a body weight scale and encouraged to weigh daily throughout the intervention and to send pictures of their weights to their interventionist before each intervention session. With very few exceptions, the first intervention session was conducted in person. Due to the COVID-19 pandemic, almost all subsequent sessions were conducted virtually via webinar (Microsoft Teams).

**Table 6 | Change in intervention satisfaction (post-intervention) in those assigned to a diet concordant vs. discordant with the genotype**

| All participants | Genotype-concordant diet (*n* = 60) Mean[a] (SD) | Genotype-discordant diet (*n* = 62) Mean[a] (SD) | Adj. difference[b] (95% CI) |
|---|---|---|---|
| I am satisfied with... | | | |
| ... the group format | 6.9 (0.4) | 7.4 (0.4) | −0.5 (−1.0, 0.1) |
| ... the support from interventionists | 7.5 (0.3) | 7.5 (0.3) | 0.1 (−0.4, 0.5) |
| ... the intervention materials | 7.0 (0.3) | 7.2 (0.3) | −0.1 (−0.6, 0.4) |
| ... the support from other participants | 6.4 (0.4) | 6.5 (0.4) | −0.1 (−0.7, 0.6) |
| ... the amount of food in my meal plan | 6.5 (0.5) | 6.4 (0.5) | 0.1 (− 0.6, 0.8) |
| ... the macronutrient content in my meal plan | 6.1 (0.4) | 5.8 (0.5) | 0.3 (−0.4, 1.0) |
| ... my progress toward weight management | 6.4 (0.5) | 6.3 (0.5) | 0.3 (−0.5, 1.0) |
| ... the degree to which the diet helped manage hunger | 6.5 (0.5) | 6.1 (0.5) | 0.5 (−0.3, 1.2) |
| **Fat-responders** | **High-fat diet (*n* = 44)** Mean[a] (SD) | **High-carbohydrate diet (*n* = 41)** Mean[a] (SD) | **Adj. difference[b] (95% CI)** |
| I am satisfied with... | | | |
| ... the group format | 6.8 (0.4) | 7.3 (0.5) | −0.6 (−1.2, 0.1) |
| ... the support from interventionists | 7.6 (0.4) | 7.8 (0.4) | −0.1 (−0.7, 0.4) |
| ... the intervention materials | 7.1 (0.4) | 7.4 (0.4) | −0.3 (−0.9, 0.3) |
| ... the support from other participants | 6.1 (0.5) | 6.5 (0.6) | −0.4 (−1.2, 0.4) |
| ... the amount of food in my meal plan | 6.3 (0.6) | 6.3 (0.6) | 0.0 (−0.9, 0.9) |
| ... the macronutrient content in my meal plan | 6.0 (0.5) | 5.6 (0.6) | 0.3 (−0.5, 1.2) |
| ... my progress toward weight management | 6.5 (0.6) | 6.2 (0.6) | 0.4 (−0.5, 1.2) |
| ... the degree to which the diet helped manage hunger | 6.4 (0.6) | 5.8 (0.6) | 0.6 (−0.3, 1.4) |
| **Carbohydrate-responders** | **High-carbohydrate diet (*n* = 16)** Mean[a] (SD) | **High-fat diet (*n* = 21)** Mean[a] (SD) | **Adj. difference[b] (95% CI)** |
| I am satisfied with... | | | |
| ... the group format | 7.2 (0.7) | 7.4 (0.7) | −0.2 (−1.3, 0.9) |
| ... the support from interventionists | 7.5 (0.6) | 6.9 (0.6) | 0.5 (−0.4, 1.4) |
| ... the intervention materials | 7.1 (0.5) | 6.6 (0.5) | 0.4 (−03, 1.1) |
| ... the support from other participants | 7.1 (0.6) | 6.4 (0.7) | 0.8 (−0.3, 1.8) |
| ... the amount of food in my meal plan | 7.0 (0.7) | 6.6 (0.8) | 0.4 (−0.8, 1.6) |
| ... the macronutrient content in my meal plan | 6.3 (0.8) | 5.9 (0.9) | 0.4 (−1.0, 1.7) |
| ... my progress toward weight management | 6.2 (0.9) | 6.4 (0.9) | −0.1 (−1.5, 1.4) |
| ... the degree to which the diet helped manage hunger | 6.8 (0.9) | 6.6 (1.0) | 0.3 (−1.3, 1.8) |

[a] Mean post-intervention value. The Intervention Satisfaction Survey was only assessed at Week 12.
[b] Adjusted for sex and race.

## Outcome measures

**Anthropometric data.** At W0 and W12, fasting body weight and waist and hip circumference were measured in the PBRC outpatient clinic. Clinic weights were also measured at all intervention visits (though not fasting weights). Further, body fat (%, via bioelectrical impedance analysis [BIA]; X-contact 365, Jawon Medical Co., Ltd, Seoul, South Korea) and blood pressure (after 5 min of seated rest) were measured at W0 and W12.

**Fasting serum glucose and insulin.** Fasting serum glucose and insulin were measured at W0, and HOMA-IR was used to quantify insulin resistance.

**Appetitive traits, food cravings, and food preferences.** Appetitive traits were measured with the Eating Inventory (EI)[46], food cravings were measured with the Food Craving Inventory (FCI)[24], and hedonic food preferences were measured with the Food Preference Questionnaire (FPQ)[47] at W0 and W12 (see Supplementary Methods for details on outcome materials). Data for these questionnaires were collected and managed using REDCap tools

**Diet personalization and intervention satisfaction.** The Diet Personalization Survey (Supplementary Methods) was completed at W0 and

W12, as well as during the intervention session at W6, and the Intervention Satisfaction Survey (Supplementary Methods) was conducted at W12. Data for these surveys were collected and managed using REDCap tools.

**Diet adherence.** As stated above, participants were provided with a kitchen scale and could precisely weigh all ingredients specified in the meal plans for the foods consumed at home. Additional foods that were consumed were weighed and added as well. Adherence to the macronutrient content of the assigned diets was assessed for three 7-day periods throughout the intervention (W4, W8, W12).

## Statistical analyses

The distribution of variables was evaluated by visual examination and the Shapiro-Wilk test. The primary outcome was weight change (kg) at 12 weeks. All other measures were secondary endpoints. Changes in outcomes are presented as mean and 95% confidence intervals (CI). We used linear mixed models to determine if changes in outcome variables differed among diets. Covariates in the models included baseline value of the outcome, sex, and race. The mixed-effect model accounted for the correlation of the subject over time, and least-square means based on the estimate from the mixed-effect model were used to test for differences in weight change between diets. To evaluate whether

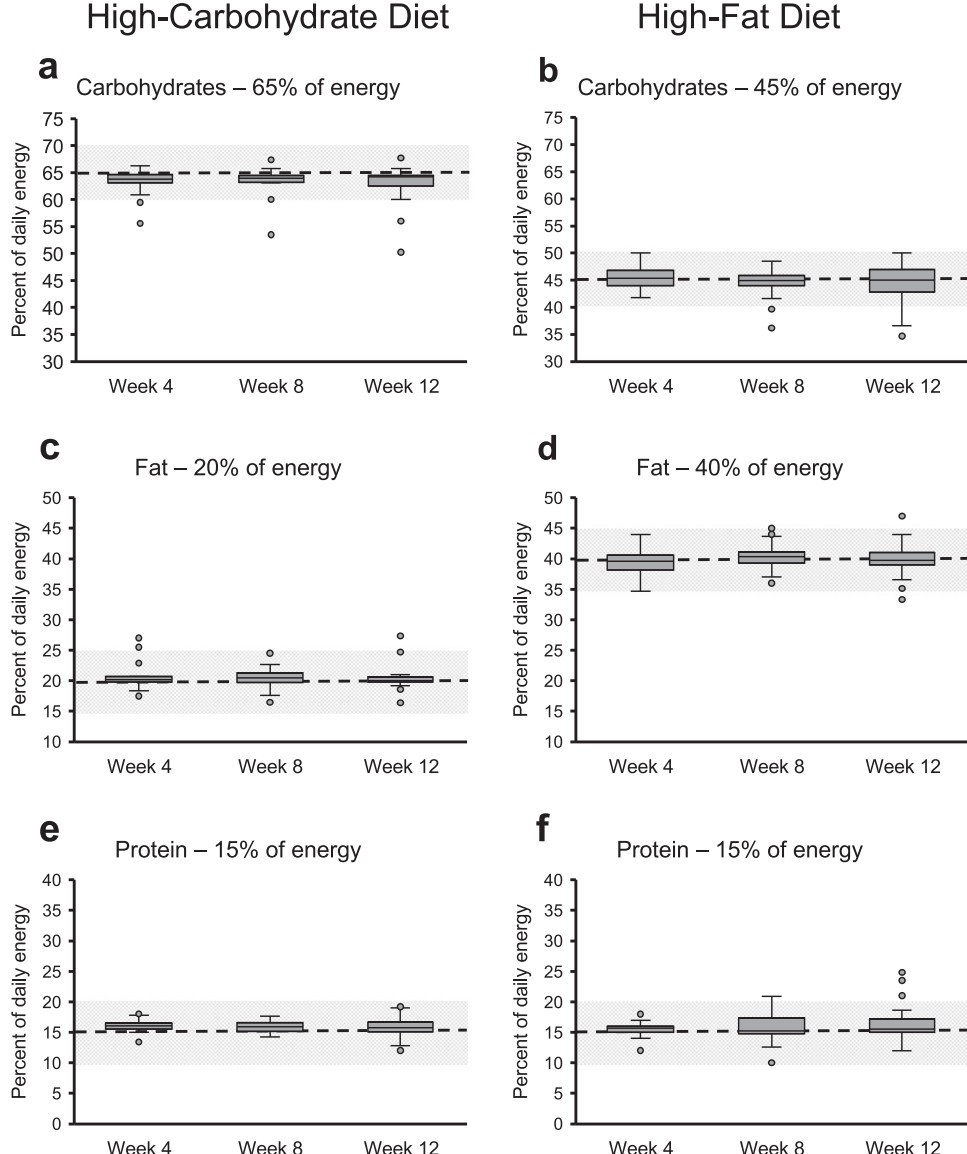

**Fig. 4 | Adherence to the macronutrient compositions of the respective diet at week 4, week 8, and week 12.** Boxplots showing adherence data for the high-carbohydrate diet (**a**, **c**, **e**) and the high-fat diet (**b**, **d**, **f**). For the high-carbohydrate diet (*n* = 22 at week 4 and 8 and *n* = 21 at week 12), target intakes were 65% carbohydrates (**a**), 20% fat (**c**), and 15% protein (**e**) and for the high-fat diet (*n* = 40 at week 4, *n* = 38 at week 8, and *n* = 37 at week 12), they were 45% carbohydrates (**b**), 40% fat (**d**), and 15% protein (**f**). The dashed line shows the target intake with the shaded area representing ±5%. In the boxplots, the center line denotes the median value (50th percentile), the bounds of the box represent the 25th and 75th percentiles of the dataset, and the whiskers mark the 5th and 95th percentiles.

baseline insulin levels and HOMA-IR needed to be included as covariates, their effects on WL were tested using a linear mixed model, adjusted for diet group and other known covariates. Neither baseline insulin levels nor HOMA-IR was significantly associated with WL; hence these variables were not included as covariates. The significance level was set to 0.05 (2-sided). Multiple testing adjustment was performed for secondary outcomes using the Holm-Bonferroni method[48]. All analyses were conducted using SAS (Windows version 9.4; SAS Institute, Cary, NC) and the statistical program R version 4.0.2 (https://cran.r-project.org/).

**Power calculations.** The present study planned to obtain data from up to 154 participants in total, and we aimed to complete 32 participants per genotype-diet group (128 participants in total) though we did not limit recruitment to achieve equal numbers of participants in each group. We hypothesized that participants on a genotype-concordant diet would lose more weight than those on a genotype-discordant diet. Based on previous studies[49,50], we assumed a standard deviation for between-group differences in weight change of 2.8 kg. To detect a 2.0 kg difference in weight change between group 1 (fat-responders on a high-fat diet) and group 2 (fat-responders on a high-carbohydrate diet) or between group 3 (carbohydrate-responders on a high-fat diet) and group 4 (carbohydrate-responders on a high-carbohydrate diet), with the intended sample size and an alpha level of 0.05, the present study would have 80% power. Further, based on the same assumptions, the present study would have >95% power to test if WL differs between participants on a genotype-concordant diet (groups 1 and 4 combined) and those on a genotype-discordant diet (groups 2 and 3).

**Reporting summary**
Further information on research design is available in the Nature Portfolio Reporting Summary linked to this article.

## Data availability

All of the data needed to recapitulate the analysis found within this study can be found in the manuscript, figures and supplementary information. Source data are provided with this paper. Due to privacy reasons, de-identified data from the study cannot be shared publicly but will be available from the corresponding author (christoph.hoechsmann@tum.de) immediately following the publication of the paper upon reasonable request. The study protocol and statistical analysis plan will also be available. Source data are provided with this paper.

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

## Acknowledgements

The center where the research was conducted is supported in part by U54 GM104940 from the National Institute of General Medical Sciences of the National Institutes of Health, which funds the Louisiana Clinical and Translational Science Center and by the Nutrition Obesity Research Centers grant P30DK072476 titled "Nutrition and Metabolic Health Through the Lifespan" sponsored by National Institute of Diabetes and Digestive and Kidney Diseases (NIDDK). C.H. was supported by an NIDDK National Research Service Award (T32DK064584), and J.L.D. was funded by the American Heart Association (Grant # 20POST35210907). The present study was funded by WW International, Inc. (New York, NY, USA). The concept and design of the study were led by the study PI (C.K.M.) and developed in collaboration with five other authors, one of whom is an employee of the sponsor. The sponsor had no role in the execution of the study or statistical analysis. Two sponsor employees are co-authors of the paper and provided editorial comments to the manuscript.

## Author contributions

C.K.M. obtained funding for the study. C.K.M., C.H., J.W.A., J.L.D., J.M.O., C.M.C., M.I.C. and G.D.F. designed the study. C.K.M. and C.H. oversaw data acquisition, and S.Y., C.H. and C.K.M. analyzed and interpreted the data. C.H. and C.K.M. drafted the manuscript; all authors provided critical revisions for important intellectual content. C.K.M. was responsible for the overall study supervision, and F.L.G. was responsible for the medical supervision.

## Funding

## Competing interests

G.D.F. and M.I.C. are shareholders and employees at WW International, Inc. (New York, NY, USA). C.K.M. has previously consulted for WW on a fee-for-service basis, with the latest consultation occurring in 2018. All other authors declare no competing interests.
