## [Peer Review File · Nature Communications]

The Personalized Nutrition Study (POINTS): Evaluation of a genetically informed weight loss approach. A Randomized Clinical TrialEditorial Note: This manuscript has been previously reviewed at another journal that is not operating a transparent peer review scheme. This document only contains reviewer comments and rebuttal letters for versions considered at *Nature Communications*. Mentions of the other journal have been redacted.

REVIEWER COMMENTS

Reviewer #1 (Remarks to the Author):

The authors have addressed my concerns in full and I have no further issues or suggestions. I am happy for the authors to see my name attached to this and my previous review.

Reviewer #2 (Remarks to the Author):

The revision is much improved.

The authors have thoroughly addressed reviewer comments.

One suggestion regarding Tables 2-5 and the presentation of excessive p-values. Collectively, across the 4 tables, there are >100 rows of variables being tested for differences between concordant vs. discordant. Given the separate columns of unadjusted and adjusted p-values, this generates >200 p-values. This appears more distracting than helpful to this reviewer.

Personal pet-peeve....p-values that go to three digits when the first digit is 0.9.

By my count there are >80 p-values listed at the >0.999 level in the far right column of Holm-Bonferroni adjustment.

I would prefer to see all the p-values removed except for the testing of primary variables. The adjusted differences and 95% CI are there for every row of data, and tell 99.9% of the story (you're right, the .9 was unnecessary).

Of the >100 unadjusted p-values, only three are below the 0.05 level and only one of those remains below that level after adjusting for multiple testing.

It seems a lot of unnecessary busyness could be taken out of the tables by removing all of

the >200 p-values other than the testing of primary outcomes.

The very few that are currently bolded could simply be mentioned, briefly, in the text.

Minor suggestions

The new figure of diet adherence is greatly appreciated, as is the acknowledgement of the high degree of missing data.

1. Could baseline values be added so that the reviewer could see and compare any differences in the magnitude of the shifts from baseline to higher fat or higher carbs?
2. Could the authors provide any explanation for the discrepancy in percent complete/missing (35% vs. 62% available data)?

Reviewer #3 (Remarks to the Author):

The authors have done a good job responding to my review of the manuscript originally submitted to [redacted].

The questions below are based on responses from the authors to the initial reviews.

- How did the biostatistician adapt the randomization process during the study? If stratification variables were added, this should be stated more explicitly.
- The "group" language is confusing in some places. Participants were randomly assigned to two groups (as indicated in the flow diagram). Stratification variables were used to ensure balanced randomization (genotype, gender, BMI), and the genotype stratification was important for testing the study hypothesis. So, there were two groups (not four groups), with two genotype strata within each group. What statistical model was tested in the analyses?
- The small number of so-called "carbohydrate-responders" is a concern. The authors acknowledge that they inaccurately estimated numbers of people with genotypically determined responsiveness to fat vs. carbohydrate when planning the study. As such, this study seems to have become an evaluation of whether high-fat or high-carbohydrate dietary interventions yield different outcomes among participants with a fat-responsive genotype.

Reviewer #4 (Remarks to the Author):

Most of my comments have been addressed very well, however some of the additions to the paper have generated a few comments from me.

Interaction p values – not sure how much value these are to the reader alone, forest plot would be useful.

What were the AEs/SAEs – just number of AEs have been stated.

“HUMA-IR did not predict WL” – associated or related to would be better than predict.

The addition of the post hoc sample size calculations should be removed. Sample size calculations should only be done to design a study not to try and interpret findings. See <https://doi.org/10.3899/jrheum.211115> and <https://doi.org/10.1198/000313001300339897>

The method of randomisation is unusual in relation to the biostatistician’s involvement – “A 1:1 allocation using random block sizes of 2, 4, and 6 was used and adapted by the biostatistician during the study to ensure equal numbers of men and women and a similar BMI across diet groups and strata.” What alterations was the statistician making to the block size? Usually this is just decided up front taking into account what the maximum and average possible imbalance could be. Why was the statistician balancing on BMI when this was not a stratification factor? What is inaction of genetic responder? Did the statistician have access to outcome data during the study when they were altering the randomisation system?

Usually, you would adjust for variables you have balanced on e.g. BMI was balanced on but not adjusted for in models. I can see why you wouldn’t adjust for this in the models for weight as you are already adjusting for baseline weight but for other outcomes no adjustment has been made.

New figures have been added. For figure 2,3,4 the mean has been plotted with the standard deviation as an error bar. The reader is likely to confuse these with confidence intervals – it

would be better to plot the data as boxplots.

We thank the reviewers for taking the time to provide a thoughtful review of our manuscript and give helpful feedback. We have addressed the remaining concerns and believe the manuscript to be much improved as a result.

Reviewer #1:

The authors have addressed my concerns in full and I have no further issues or suggestions. I am happy for the authors to see my name attached to this and my previous review.

Thank you again for making the time to provide such a thoughtful review. We are grateful.

Reviewer #2:

The revision is much improved. The authors have thoroughly addressed reviewer comments.

1. One suggestion regarding Tables 2-5 and the presentation of excessive p-values. Collectively, across the 4 tables, there are >100 rows of variables being tested for differences between concordant vs. discordant. Given the separate columns of unadjusted and adjusted p-values, this generates >200 p-values. This appears more distracting than helpful to this reviewer. Personal pet peeve....p-values that go to three digits when the first digit is 0.9. By my count, there are >80 p-values listed at the >0.999 level in the far right column of Holm-Bonferroni adjustment. I would prefer to see all the p-values removed except for the testing of primary variables. The adjusted differences and 95% CI are there for every row of data, and tell 99.9% of the story (you're right, the .9 was unnecessary). Of the >100 unadjusted p-values, only three are below the 0.05 level and only one of those remains below that level after adjusting for multiple testing. It seems a lot of unnecessary busyness could be taken out of the tables by removing all of the >200 p-values other than the testing of primary outcomes. The very few that are currently bolded could simply be mentioned, briefly, in the text.

We agree that the presentation of this many p-values created unnecessary busyness in the tables. As suggested, we have removed all p-values from Tables 2-5 (except for the primary outcome) as well as from all tables in the supplement, and only mention those below 0.05 briefly in the text.

Minor suggestions

The new figure of diet adherence is greatly appreciated, as is the acknowledgement of the high degree of missing data.

1. Could baseline values be added so that the reviewer could see and compare any differences in the magnitude of the shifts from baseline to higher fat or higher carbs?

Unfortunately, we did not assess the macronutrient composition of the habitual diet before the intervention and we only assessed adherence to the assigned diet during the 3 time periods (week 4, 8, and 12). During the first week, the interventionists were working closely with the participants to make sure the assigned meal plans were followed as precisely as possible, though adherence was not "measured" during that week in same way as in week 4, 8, and 12. We noted this point in the study limitations:

"In addition to assessing diet adherence continuously throughout the study, future studies should also assess the macronutrient composition of participants' habitual diets to see any differences in the magnitude of the shifts from baseline to the high-fat or high-carbohydrate diet."

2. Could the authors provide any explanation for the discrepancy in percent complete/missing (35% vs. 62% available data)?

We cannot explain the difference, though we have no reason to believe that it was systematic. We acknowledged this discrepancy in the Results section and added the following:

“The discrepancy in the percent complete/missing is noted, though we have no reason to believe that it was systematic.”

Reviewer #3:

The authors have done a good job responding to my review of the manuscript originally submitted to [redacted]. The questions below are based on responses from the authors to the initial reviews.

1. How did the biostatistician adapt the randomization process during the study? If stratification variables were added, this should be stated more explicitly.

The biostatistician created an equation that was developed before the start of enrolment, and it was only edited to indicate the block sizes for each gender. The equation was designed in such a way that it took into account the group BMI means and the current BMI of the enrolled participant. BMI was treated as a continuous variable, as compared to a categorical version, to better account for its effect. The randomization section is edited below (see Reviewer #4’s comment #5) to better describe the methods.

2. The "group" language is confusing in some places. Participants were randomly assigned to two groups (as indicated in the flow diagram). Stratification variables were used to ensure balanced randomization (genotype, gender, BMI), and the genotype stratification was important for testing the study hypothesis. So, there were two groups (not four groups), with two genotype strata within each group. What statistical model was tested in the analyses?

Thank you for pointing this out. You are correct that the group language was confusing. Participants were identified *a priori* as carbohydrate-responders and fat-responders and randomized to either a high-carbohydrate or high-fat diet, yielding the 4 specified genotype-diet groups.

	High-fat diet	High-carbohydrate diet
Fat-responder	A	B
Carbohydrate responder	C	D

We used linear mixed models to determine if changes in outcome variables differed among genotype-concordant (cells A+D) and genotype discordant diets (cell B+C) for the whole sample as well as between the high-fat diet (cell A) and high-carbohydrate diet (cell B) among fat-responders and between the high-carbohydrate diet (cell D) and high-fat diet (cell C) among carbohydrate responders. We clarified this throughout the text.

3. The small number of so-called "carbohydrate-responders" is a concern. The authors acknowledge that they inaccurately estimated numbers of people with genotypically determined responsiveness to fat vs. carbohydrate when planning the study. As such, this study seems to have become an evaluation of whether high-fat or high-carbohydrate dietary interventions yield different outcomes among participants with a fat-responsive genotype.

Interesting points. The study was designed to randomize people as they qualified; hence, we did not *a priori* plan to recruit an equal number of carbohydrate- vs. fat-responders, though it was implied since we originally assumed that an equal number of carbohydrate- and fat-responders were present in the population. We think it is fair to say that we were better powered to evaluate whether high-fat or high-carbohydrate dietary interventions yield different outcomes among participants with a fat-responsive genotype, though we believe that we already noted this, and we did include the post-hoc power calculations in the Discussion. Please note, that Reviewer #4 requested those be removed, however.

Reviewer #4:

Most of my comments have been addressed very well, however some of the additions to the paper have generated a few comments from me.

1. Interaction p values – not sure how much value these are to the reader alone, forest plot would be useful.

We have added a forest plot to the supplement (Supplementary Figure 2), as suggested.

2. What were the AEs/SAEs – just number of AEs have been stated.

There were 4 AEs/SAEs in total, with 2 AEs and 2 SAEs. The 2 AEs occurred among fat-responders on a high-carbohydrate diet: (1) lymphadenopathy – caused by lymphedema that was diagnosed after a surgery that occurred >1 year before study and (2) lipedema and fluid retention following an ankle surgery >2 years before study. As to the 2 serious adverse events, 1 occurred among fat-responders on a high-carbohydrate diet (colon abscess; history of) and 1 among fat-responders on a high-fat diet (colon abscess; history of diverticulitis RSV infection), with both SAEs requiring hospitalization. None of the 4 AEs/SAEs were likely related to the study. We have added the following to the respective section in the manuscript:

“There were 4 adverse or serious adverse events in total. Two adverse events occurred among fat-responders on a high-carbohydrate diet (unrelated to study), and there were 2 serious adverse events (1 among fat-responders on a high-carbohydrate diet, 1 among fat-responders on a high-fat diet) that required hospitalization (unrelated to study).”

3. “HOMA-IR did not predict WL” – associated or related to would be better than predict.

Changed as suggested.

4. The addition of the post hoc sample size calculations should be removed. Sample size calculations should only be done to design a study not to try and interpret findings. See <https://doi.org/10.3899/jrheum.211115> and <https://doi.org/10.1198/000313001300339897>

We removed the post hoc sample size calculations from the Discussion, as suggested.

5. The method of randomisation is unusual in relation to the biostatistician's involvement – “A 1:1 allocation using random block sizes of 2, 4, and 6 was used and adapted by the biostatistician during the study to ensure equal numbers of men and women and a similar BMI across diet groups and strata.” What alterations was the statistician making to the block size? Usually this is just decided up front taking into account what the maximum and average possible imbalance could be. Why was the statistician balancing on BMI when this was not a stratification factor? What is inaction of genetic responder? Did the statistician have access to outcome data during the study when the were altering the randomisation system?

The biostatistician was not balancing on BMI, as the randomization equation was handling that aspect of the process (see Reviewer #3's comment #1). The biostatistician only adjusted the current block sizes. Block sizes of 6 were the default for females while block sizes of 4 were used for males. That decision was made due to issues in recruiting male subjects in previous studies (leading to >70% female participants in these studies, e.g., Tate et al., 2022, DOI: 10.1001/jamanetworkopen.2022.26561 and Apolzan et al., 2023, DOI: 10.1038/s41387-023-00234-6) and to keep diet groups random but balanced on sex. Once the study reached 85% of the enrolment goal, the block sizes were reduced to 4 and 2, respectively, to ensure relative balance between groups within each stratum. As described in our response to Reviewer #3's comment #1, BMI was treated as a continuous variable and therefore could not be used a stratification factor. The randomization equation was kept separate from any outcome related data. We added the following paragraph to the Methods section:

“To ensure a relatively equal baseline BMI between the 4 genotype-diet groups, a 1:1 randomization scheme was devised that adjusted for BMI, gender, and genotype. Gender and genotype were used as strata while BMI was used in an a-priori-created randomization equation. This equation, within each stratum, used block sizes of 6 (for females) and 4 (for males) at the start of the study and ended with block sizes of 4 and 2, respectively, to ensure relative balance of group assignments. Block sizes were assigned during the study by the biostatistician with access only to information about the enrolment progress (percent enrolled).”

6. Usually, you would adjust for variables you have balanced on e.g. BMI was balanced on but not adjusted for in models. I can see why you wouldn't adjust for this in the models for weight as you are already adjusting for baseline weight but for other outcomes no adjustment has been made.

We did not adjust for BMI in the models, following the statistical analyses plan as specified in the protocol. However, per your request, we additionally ran all models with adjustment for BMI, and the results do not differ meaningfully from those without additional BMI adjustment. Therefore, we prefer to present the results without adjustment.

7. New figures have been added. For figure 2,3,4 the mean has been plotted with the standard deviation as an error bar. The reader is likely to confuse these with confidence intervals – it would be better to plot the data as boxplots.

We plotted the data as boxplots in these figures, as suggested.

REVIEWERS' COMMENTS

Reviewer #2 (Remarks to the Author):

The authors have satisfactorily addressed my concerns. I have no further comments.

Reviewer #3 (Remarks to the Author):

The authors have addressed my comments.

Reviewer #4 (Remarks to the Author):

Comments are well addressed. One minor thing regarding the forest plot - I think the readers would find it easier to interpret if it was done with the main effect and each of the interaction effects combined and the reference line on the overall treatment effect for the study (rather than at 0). See [https://doi.org/10.1016/S0140-6736\(05\)61026-4](https://doi.org/10.1016/S0140-6736(05)61026-4) for an explanation and example.

REVIEWERS' COMMENTS

Reviewer #2 (Remarks to the Author):

The authors have satisfactorily addressed my concerns. I have no further comments.

Reviewer #3 (Remarks to the Author):

The authors have addressed my comments.

Reviewer #4 (Remarks to the Author):

Comments are well addressed. One minor thing regarding the forest plot - I think the readers would find it easier to interpret if it was done with the main effect and each of the interaction effects combined and the reference line on the overall treatment effect for the study (rather than at 0). See [https://doi.org/10.1016/S0140-6736\(05\)61026-4](https://doi.org/10.1016/S0140-6736(05)61026-4) for an explanation and example.

Thank you for this suggestion. We have edited the forest plot as suggested.